https://doi.org/10.1038/s41467-019-10220-1　　**OPEN**

# The UbiX flavin prenyltransferase reaction mechanism resembles class I terpene cyclase chemistry

Stephen A. Marshall [1], Karl A.P. Payne [1], Karl Fisher [1], Mark D. White[1,2], Aisling Ní Cheallaigh[1,3], Arune Balaikaite[1], Stephen E.J. Rigby[1] & David Leys[1]

The UbiX-UbiD enzymes are widespread in microbes, acting in concert to decarboxylate alpha-beta unsaturated carboxylic acids using a highly modified flavin cofactor, prenylated FMN (prFMN). UbiX serves as the flavin prenyltransferase, extending the isoalloxazine ring system with a fourth non-aromatic ring, derived from sequential linkage between a dimethylallyl moiety and the FMN N5 and C6. Using structure determination and solution studies of both dimethylallyl monophosphate (DMAP) and dimethyallyl pyrophosphate (DMAPP) dependent UbiX enzymes, we reveal the first step, N5-C1' bond formation, is contingent on the presence of a dimethylallyl substrate moiety. Hence, an $S_N1$ mechanism similar to other prenyltransferases is proposed. Selected variants of the (pyro)phosphate binding site are unable to catalyse subsequent Friedel-Crafts alkylation of the flavin C6, but can be rescued by addition of (pyro)phosphate. Thus, retention of the (pyro)phosphate leaving group is required for C6-C3' bond formation, resembling pyrophosphate initiated class I terpene cyclase reaction chemistry.

[1] Manchester Institute of Biotechnology, University of Manchester, Manchester M1 7DN, UK. [2] Present address: School of Chemistry, The University of Sydney, Sydney NSW 2006 New South Wales, Australia. [3] Present address: Centre for Synthesis and Chemical Biology, University College Dublin, Dublin D04 V1W8, Ireland. Correspondence and requests for materials should be addressed to D.L. (email: david.leys@manchester.ac.uk)

The UbiD–UbiX decarboxylase system is primarily known for its role in the production of quinone cofactors[1]. It is most commonly associated with the ubiquinone biosynthetic pathway (from which it receives its name), catalysing the decarboxylation of 3-octaprenyl-4-hydroxybenzoic acid[2]; however, homologues have been shown to act in other quinone biosynthesis, including plastoquinone[3]. Recently, the individual roles of UbiD and UbiX have been discovered[4,5]. Originally thought to be redundant but distinct decarboxylases, it has been shown that UbiD is in fact the only decarboxylase, whereas UbiX produces a cofactor strictly required for UbiD activity. However, UbiD–UbiX homologues are also involved in the decarboxylation of a wide variety of enoic and aromatic acid substrates[6]. Furthermore, there are examples of UbiD–UbiX involvement in the production of secondary metabolites[7,8], as exemplified by the TtnD–TtnC system involved in the production of tautomycetin, which is an antifungal and anticancer compound[9].

The decarboxylation catalysed by UbiD presents a reaction of biotechnological interest, but the reverse reaction (carboxylation) can also be catalysed by UbiD enzymes under high $CO_2$ concentrations, allowing for the production of synthetically useful acid compounds. Indeed, it has recently been demonstrated that furan dicarboxylic acid (FDCA), a potential green replacement for PET plastics, can be produced by the UbiD-driven carboxylation of furoic acid[10]. Furoic acid is an abundant waste chemical from pre-treatment of lignocellulosic material in the production of biofuels[11,12], and therefore represents an ideal feedstock for the production of bioplastics. In fact, it is proposed that UbiD–UbiX mediated carboxylation allows specialised microbes to degrade benzene[13], phenol[14] or polyaromatic compounds under anaerobic conditions[15]. Hence, the natural variety of substrates which can be (de)carboxylated by the ubiquitous UbiD decarboxylase family renders the system an ideal target for future applications. Given the key role UbiX plays, a full understanding of its mechanism is required to provide a blueprint for future application and engineering of this system.

The discovery that UbiD enzymes depend on the unusual cofactor prenylated FMN (prFMN) for activity, paved the way for determining the role of UbiX[16,17]. It was shown that the latter catalyses a flavin prenyltransferase reaction, using reduced $FMNH_2$ and dimethylallyl monophosphate (DMAP) as substrates to form $prFMNH_2$ (i.e., reduced prFMN)[5]. The $prFMNH_2$ product frequently remains bound to UbiX in vitro. In the presence of oxygen, UbiX-bound $prFMNH_2$ readily oxidises to form a stable and purple coloured prFMN· semiquinone radical species. It is proposed that in vivo $prFMNH_2$ is transferred to the UbiD partner, which in turn catalyses the oxidative maturation to the active $prFMN^{iminium}$ form (see refs. [4,18]; Fig. 1). The iminium form of prFMN has azomethine ylide character[19], believed to support an unusual 1,3-dipolar cycloaddition reaction with the alpha–beta unsaturated carboxylic acid substrate[20,21], which ultimately leads to decarboxylation[4,6,9]. Recent studies revealed some UbiX homologues prefer the canonical dimethylallylpyrophosphate isoprenoid precursor (DMAPP)[22], as opposed to the unusual DMAP preference reported for the *Pseudomonas aeruginosa* UbiX (*Pa*UbiX[5]) and other UbiX enzymes[23]. Crystal structures of *Pa*UbiX in complex with substrates reveal a highly conserved phosphate-binding motif that places the substrate within a hydrophobic pocket consisting of the flavin isoalloxazine ring and various aromatic residues. Kinetic crystallography revealed that *Pa*UbiX first catalyses the N5–C1′ bond formation, leading to an N5-prenyl-FMN species, followed by the Friedel–Crafts alkylation of the flavin C6 ([5]; Fig. 1). The phosphate-leaving group is proposed to assist with the latter process.

It is unclear what governs the respective DMAP/DMAPP specificity of individual UbiX enzymes, and whether this is coupled to any divergence in the UbiX mechanism in view of the key role suggested for the (pyro)phosphate-leaving group in the C6–C3′ bond formation.

In order to understand the DMAP versus DMAPP preference and any mechanistic consequences, we here determine the crystal structure of the DMAPP-specific *Aspergillus niger* UbiX (*An*UbiX; also called PadA1). This shows that the intricate UbiX:alpha-phosphate interaction network is conserved with the DMAP-specific *Pa*UbiX, while additional space is available to accommodate the additional DMAPP beta-phosphate group. Using several UbiX homologues as well as selected *Pa*UbiX variants, in combination with DMAP(P) and DMAP-like substrates, we reveal the first step, N5–C1′ bond formation (i.e., the prenyltransferase step), likely occurs through an $S_N1$ mechanism. This leads to the strict requirement for a substrate dimethylallyl moiety to initiate the overall reaction. We also reveal that the N5 alkylation process occurs when using DMAP in the DMAPP-specific *An*UbiX, and is thus independent of the presence of the DMAPP beta-phosphate. However, in this case we show the ensuing Friedel–Crafts alkylation of the flavin C6 only occurs when using DMAPP. Selected *Pa*UbiX variants of the phosphate-binding site are also unable to catalyse C6–C3′ bond formation, but can be rescued by addition of phosphate. This confirms UbiX catalyses the C6–C3′ bond formation through (pyro)phosphate acid–base catalysis, similar to what has been proposed for class I terpene cyclases[24,25].

**Fig. 1** UbiX and UbiD act in concert to produce $prFMN^{iminium}$. Proposed mechanism for flavin prenyltransferase activity by UbiX, followed by transfer of $prFMNH_2$ to UbiD and subsequent oxidative maturation to the active $prFMN^{iminium}$ cofactor

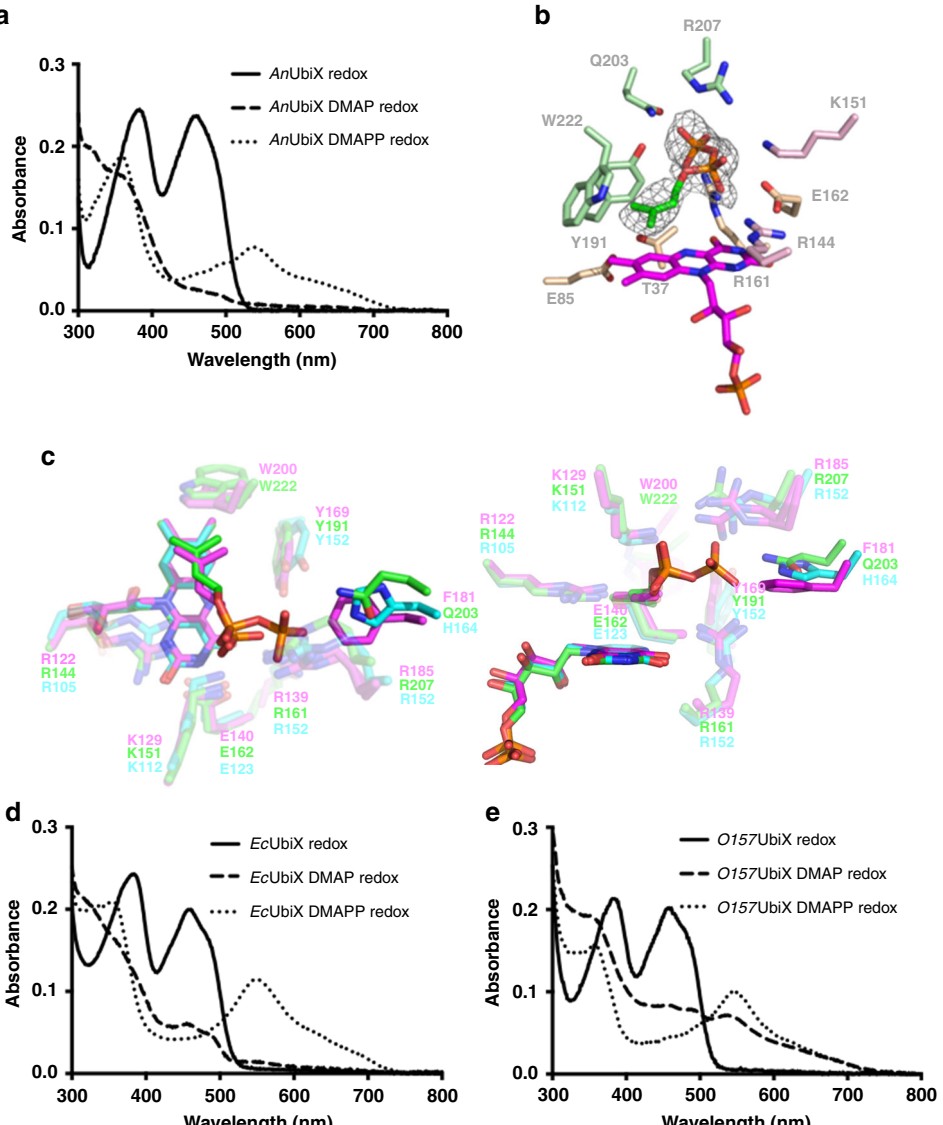

**Fig. 2** Structure and properties of DMAPP-specific UbiX enzymes. **a** UV–Vis spectra of FMNH$_2$:*An*UbiX incubated with buffer, DMAP or DMAPP, respectively, following oxidation. Only the DMAPP substrate leads to formation of a purple prFMN semiquinone. Crucially, DMAP incubation does affect the resulting FMN spectrum. **b** The crystal structure of *An*UbiX in complex with FMN and DMAPP. The *An*UbiX active site is shown with the omit polder map corresponding to bound DMAPP contoured at 3 sigma. Residues are coloured according to different *An*UbiX monomers. **c** A structural overlay of *Pa*UbiX (magenta), *An*UbiX (green) and *O157*UbiX (cyan) active sites. This reveals a highly conserved network of residues with the exception of divergence at position 181 (*Pa*UbiX numbering). Right image is a 90° rotation about the *x*-axis. **d** UV–Vis spectra of FMNH$_2$:*Ec*UbiX incubated with buffer, DMAP or DMAPP, respectively, following oxidation. A similar behaviour to *An*UbiX is observed. **e** UV–Vis spectra of FMNH$_2$:*O157*UbiX incubated with buffer, DMAP or DMAPP, respectively, following oxidation. Formation of purple prFMN semiquinone is observed both with DMAP and DMAPP, although the yield is higher with DMAPP. Stereo images of b and c are available in Supplementary Figs. 1 and 2, respectively

## Results

**Structural basis for UbiX DMAP/DMAPP specificity**. Following the discovery that the *Saccharomyces cerevisiae* UbiX (also called Pad1) is capable of flavin prenyltransferase activity with DMAPP[22], we confirmed that the highly similar (61% identity, 78% similarity—excluding the mitochondrial targeting region) *Aspergillus niger* UbiX (*An*UbiX) also requires DMAPP as opposed to DMAP in order to produce prFMNH$_2$ (Fig. 2a). We determined the crystal structure of *An*UbiX to determine the structural basis of DMAPP versus DMAP specificity. Although *An*UbiX was crystallised without any exogenous addition of DMAPP, the crystal structure reveals electron density directly located above the oxidised FMN that corresponds to a bound DMAPP (Fig. 2b). The DMAPP:*An*UbiX complex closely

resembles the DMAP:*Pa*UbiX complex[5], and the DMAPP alpha-phosphate is bound by *An*UbiX in a manner nearly identical to the *Pa*UbiX DMAP alpha-phosphate interaction. While the majority of the active-site residues responsible for DMAP(P) binding are conserved in *An*UbiX/*Pa*UbiX, a significant variation occurs at position 181 (*Pa*UbiX numbering). In *Pa*UbiX, F181 appears to block binding of a beta-phosphate containing DMAPP in *Pa*UbiX. In comparison, a Gln residue is present in *An*UbiX at this position (Q203), establishing a hydrogen bond interaction with the DMAPP beta-phosphate (Fig. 2c).

We extended our studies to include UbiX from *E. coli* K12 (*Ec*UbiX) and *E. coli* O157:H7 (for which a crystal structure is available, PDB 1SBZ;[26] *O157*UbiX). The *Ec*UbiX substrate preference resembles *An*UbiX, and is only able to catalyse

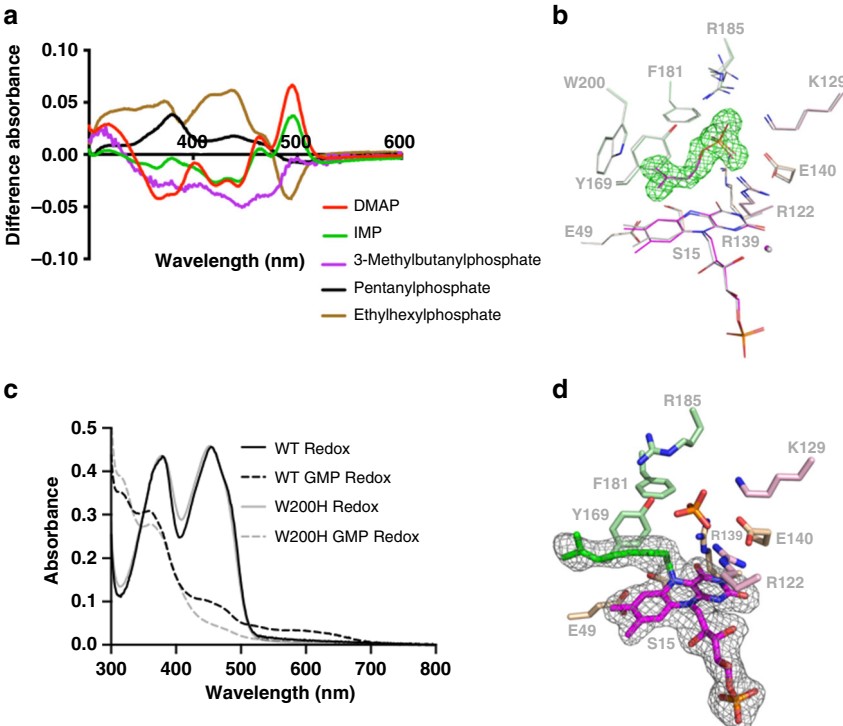

**Fig. 3** The N5-prenylation step in UbiX occurs via $S_N1$. **a** Difference spectra following addition of various DMAP-like alkyl-phosphate compounds with WT *Pa*UbiX, indicative of binding in close proximity of the FMN by all the compounds tested. Structures of compounds can be found in Supplementary Fig 3. **b** The crystal structure of the *Pa*UbiX: $FMNH_2$:IMP complex superposed with previously determined *Pa*UbiX:FMN:DMAP (4ZAF) in grey. The omit polder map corresponding to bound IMP is shown in green mesh contoured at 5 sigma. Residues are coloured according to *Pa*UbiX monomer. **c** UV–Vis spectra following oxidation of $FMNH_2$ bound WT *Pa*UbiX and the W200H variant incubated with geranyl monophosphate. **d** The crystal structure of *Pa*UbiX W200H with an N5 geranyl adduct (green) following incubation of reduced crystals with geranyl monophosphate. Omit polder map corresponding to the bound geranyl N5 FMN shown in grey mesh contoured at 4 sigma. The W200H mutation is not visible due to disorder of the C-terminus. Stereo images of **b** and **d** are available in Supplementary Figs. 4 and 5, respectively

DMAPP-specific $prFMNH_2$ formation under the conditions tested. In contrast, *O157*UbiX is able to use both DMAP and DMAPP to generate $prFMNH_2$ (Fig. 2d, e). This clearly confirms DMAP/DMAPP specificity does not correlate with prokaryotic versus eukaryotic origin of the UbiX. The structure of *O157*UbiX was previously solved with FMN bound in the active site[26]. Although not present in the deposited model, electron density corresponding to a bound phosphate or sulphate ion occupying the conserved alpha-phosphate-binding site appears present. A structural overlay of *O157*UbiX with the *Pa*UbiX and *An*UbiX crystal structures shows that the equivalent residue to *Pa*UbiX F181 is *O157*UbiX H164. The latter is likely to form a hydrogen bond with the beta-phosphate similar to *An*UbiX Q203 (Fig. 2c), and thus explains why *O157*UbiX is capable of performing catalysis with DMAPP.

In view of the highly similar DMAPP/DMAP binding, it is unclear why *An*UbiX (and by extension *Ec*UbiX) is unable to catalyse DMAP-dependent prFMN formation. This suggests a very specific role for the pyrophosphate-leaving group in catalysis. On the other hand, the observation *O157*UbiX can catalyse the prenyltransferase step with both DMAP and DMAPP, suggesting that the beta-phosphate does not play an essential role.

### The prenyltransferase step occurs via an $S_N1$ mechanism.
*Ec*UbiX and *An*UbiX do not generate a prFMN• purple radical species when undergoing consecutive reduction and oxidation processes in presence of DMAP. However, at the end of the redox

cycle, the UV–Vis spectrum of the mixture is perturbed compared with that of the starting state prior to the reaction, and does not resemble that of oxidised FMN (Fig. 2a, d). The formation of a distinct spectral species following the redox cycle suggests a reaction between DMAP and $FMNH_2$ has occurred in both *Ec*UbiX and *An*UbiX, but that the resulting adduct did not proceed to form $prFMNH_2$ (given no purple prFMN• Semiquinone formation was observed following oxidation; Fig. 2a, d). We propose this adduct corresponds to the N5-prenylated FMN species. To further investigate the N5-prenylation step, we determined the reactivity of *Pa*UbiX with a series of monophosphorylated compounds resembling DMAP.

Perturbation of the oxidised *Pa*UbiX-bound FMN spectrum indicates a wide range of DMAP-like compounds bind in the *Pa*UbiX active site (Fig. 3a; Supplementary Fig 3). However, redox cycling of *Pa*UbiX in the presence of these ligands did not alter the flavin spectrum, indicating no FMN modification has occurred. Interestingly, even the DMAP isomer, isopentenyl monophosphate (IMP), is unable to initiate any flavin alkylation reaction. We determined the *Pa*UbiX:$FMNH_2$:IMP ternary complex crystal structure, resembling the catalytically active *Pa*UbiX:$FMNH_2$:DMAP complex (Fig. 3b). This reveals IMP is indeed bound in a nearly identical manner to DMAP, located directly above the $FMNH_2$ isoalloxazine plane. A comparison with the previously determined *Pa*UbiX:FMN:DMAP complex structure (4ZAF) reveals flavin reduction is accompanied by a distortion of the isoalloxazine plane (from 173.2° to 161.5° across N5), while shortening the N5–$C1^{IMP}$ distance to 3.17 Å (as compared with the 3.33 Å N5–$C1^{DMAP}$ distance). FMN reduction

is also accompanied by a minor decrease in the distance between FMN N1 and a water molecule. Crucially, no formation of an N5–C1′ bond can be observed, a process that occurs rapidly upon reduction of PaUbiX:FMN:DMAP crystals[5]. Thus, a dimethylallyl moiety is essential for reactivity, as neither saturated DMAP-like compounds nor the isomer IMP lead to N5 alkylation or prFMN formation.

To further validate this hypothesis, we used geranyl monophosphate (GMP) as an alternative substrate (Supplementary Fig. 3). This compound contains the required dimethylallyl moiety, but is twice the size of the DMAP substrate. PaUbiX appears to be able to form a geranyl N5–C1′ linkage, by virtue of the change in the UV–Vis spectrum upon redox cycling in the presence of GMP (Fig. 3c). However, GMP binding could not be detected in PaUbiX crystals, which we suggest is due to the presence of W200 that likely blocks GMP access to the active site in crystallo. Indeed, crystals of a PaUbiX W200H variant were soaked with GMP and subjected to reducing conditions to initiate the reaction. The corresponding electron density obtained from reduced PaUbiX W200H:GMP crystals confirms formation of an N5–C1′ bond with the geranyl group (Fig. 3d).

**Phosphate-binding pocket mutations perturb C6-alkylation**. In an attempt to further probe the DMAP/DMAPP specificity, and explore the putative role of the (pyro)phosphate in Friedel–Crafts C6-alkylation, we constructed F181Q (to mimic AnUbiX) and F181H (to mimic O157UbiX) PaUbiX variants. In each variant, the DMAP specificity was retained, and crystal structures of the corresponding complexes with oxidised FMN and DMAP revealed little difference with the WT PaUbiX structure besides the mutations itself (Fig. 4a, c). This clearly demonstrates the nature of the residue at position 181 is not the sole determinant of DMAP/DMAPP specificity. Upon reduction of the PaUbiX variant crystals, it was apparent that F181H was able to catalyse prFMN formation (Fig. 4b). In contrast, the F181Q mutant does not proceed beyond the N5-prenylated intermediate (Fig. 4d). We have previously identified a similar intermediate species in both PaUbiX Y169F and E49Q crystal structures (4ZAY, 4ZAZ[5]).

The in crystallo behaviour of both variants is mirrored in solution: F181H is able to form the purple species indicative of the prFMN· semiquinone, while redox cycling of F181Q with DMAP leads to a spectral species with features reminiscent of the DMAP reactions with the DMAPP-specific AnUbiX/EcUbiX (Fig. 4e, f). Using similar conditions, we here show that the UV–Vis spectrum of redox cycled PaUbiX Y169F and E49Q in presence of DMAP is also similar (Fig. 4g). Following reoxidation of PaUbiX F181Q under single turnover conditions, the corresponding EPR spectrum is a single isotropic line with a peak to trough width of 18.3 G at $g = 2.0033$ (Fig. 4h). This signal lacks the partially resolved hyperfine structure observed previously for prFMN· and is consistent with an FMN neutral (blue) semiquinone species. Given that the orientation of the C1′-protons relative to the flavin N5 in the N5-prenylated intermediate is very different to that observed for prFMN, the contribution of the C1′-protons to the EPR spectrum would be expected to be minimal and thus we cannot exclude the possibility that this corresponds to a N5-prenylated intermediate semiquinone species.

It is proposed that the conserved E49 assists Friedel–Crafts alkylation by proton abstraction from the isoalloxazine C6[5]. The PaUbiX E49Q crystal structure reveals an N5-alkylated adduct that appears to have undergone a C2′–C3′ isomerization, which is not observed in the Y169F or F181Q variants (Supplementary Fig. 10). The only group that is a likely candidate for catalysing acid/base-mediated isomerization of the N5-bound dimethylallyl

moiety is the phosphate group, and this process appears compromised in PaUbiX Y169F/F181Q variants. A perturbation of the phosphate $pK_a$ and/or its position relative to the dimethylallyl moiety could prevent formation of the allylic carbocation following C2′–C3′ isomerisation. In a similar way, reaction of the strict DMAPP-specific UbiX enzymes with DMAP might lead to incorrectly positioned phosphate, arresting the reaction at the N5-prenyl stage. Alternatively, the relatively slow C6-alkylation step is hindered by the loss of the weakly bound phosphate in these enzymes prior to the C2′ protonation step. The crystal structures of both Y169F and F181Q retain the phosphate-leaving group in the redox cycled state, despite being unable to form the C3′–C6 bond. However, at the resolution obtained, we are unable to unequivocally state whether the position of the phosphate group is perturbed relative to the C2′ of the DMAP adduct, and it may be the case that the rigidity of the active site in the crystal prevents the total loss of the phosphate group.

**(Pyro)phosphate addition salvages the C6–C3′ bond formation**. To explore whether the loss of phosphate is indeed responsible for the lack of DMAP-dependent prFMN formation in DMAPP-specific enzymes, we attempted to salvage prFMN formation from the stalled N5-prenyl adduct by the addition of 5 mM (pyro)phosphate to the reaction. We show that in the case of EcUbiX, the addition of either phosphate or pyrophosphate is sufficient to allow DMAP-dependent production of prFMN under single turnover conditions, as demonstrated by formation of the prFMN• radical upon oxidation. Crucially, addition of sulphate did not salvage prFMN formation (Fig. 5a). Furthermore, we tested whether premature phosphate loss could likewise explain the PaUbiX Y169F/F181Q properties. Indeed, addition of 5 mM phosphate to the stalled N5-prenylated Y169F/F181Q species prior to oxidation leads to formation of the purple prFMN. This is further confirmed by the change in the EPR spectrum of the oxidised N5-adduct F181Q species when phosphate is added. Upon addition of 5 mM phosphate, the EPR spectrum returns to a characteristic prFMN•, confirming the Friedel–Crafts C6-alkylation step can occur for the semiquinone species and is phosphate dependent (Fig. 4h). In contrast, product formation in the E49Q variant is not salvaged by the addition of phosphate, indicative of the distinct effect this mutation has on catalysis (Fig. 5b).

Hence, it appears that (pyro)phosphate affinity is key to ensuring that the (pyro)phosphate leaving group from the prenyltransferase $S_N1$ reaction is retained, in order for it to catalyse the slow C6-alkylation step by protonation of the dimethylallyl C2′. Supplementing the reaction with phosphate enables both DMAPP-specific enzymes as well as PaUbiX variant enzymes to proceed past the otherwise stalled N5-prenylated species.

**Discussion**
The UbiX enzyme catalyses both the N5–C1′ and C6–C3′ bond formation to generate the fourth ring of the prFMN product (Fig. 6). The reduction of the flavin substrate is strictly required to initiate the N5–C1′ bond formation step, and either is required for an $S_N2$ type attack on the DMAP(P) substrate or, in the case of an $S_N1$ mechanism, to assist formation of the DMA carbocation through pi–pi stacking. An $S_N2$ mechanism can also, in principle, occur with other alkylated phosphates. The lack of activity with all but DMAP(P) substrate (or GMP) confirms the first step is an $S_N1$-type prenyltransferase reaction dependent on formation of a dimethylallyl-derived resonance-stabilised carbocation. This is similar to that proposed for other evolutionary

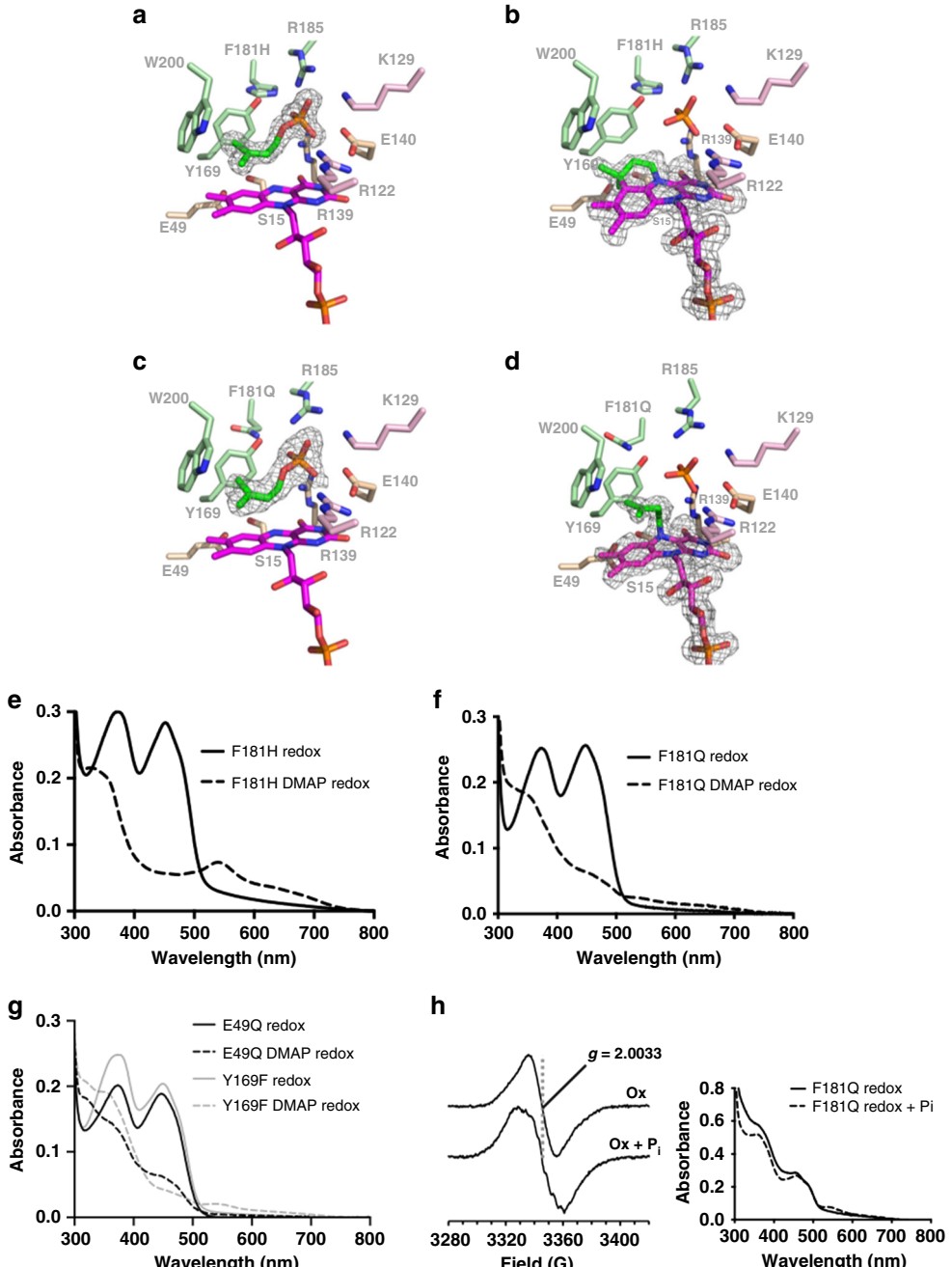

**Fig. 4** Phosphate retention is key to C6–C3' bond formation in *Pa*UbiX. **a** The crystal structure of the F181H *Pa*UbiX variant in complex with DMAP and FMN. **b** The crystal structure of F181H *Pa*UbiX variant crystals following reduction and subsequent reoxidation revealing formation of prFMN. **c** The crystal structure of the F181Q *Pa*UbiX variant in complex with DMAP and FMN. **d** The crystal structure of F181Q *Pa*UbiX variant crystals following reduction and subsequent reoxidation revealing the reaction does not proceed beyond N5-prenylation in this variant. Omit polder maps corresponding to bound DMAP or FMN adducts are contoured at 3 sigma. **e** UV–Vis spectra of FMNH₂:F181H *Pa*UbiX incubated with respectively buffer or DMAP following oxidation. **f** UV–Vis spectra of FMNH₂:F181Q *Pa*UbiX incubated with buffer or DMAP, respectively, following oxidation. The lack of any features at 550 nm is indicative of the formation of a stalled N5-C15-C1 at 3. **g** UV–Vis spectra of FMNH₂:Y169F and E49Q *Pa*UbiX variants incubated with buffer or DMAP, respectively, following oxidation. **h** EPR spectra of FMNH₂:F181Q *Pa*UbiX following incubation with DMAP and subsequent oxidation, and following addition of phosphate after oxidation. In the absence of phosphate supplementation, the spectrum is consistent with a blue FMN semiquinone species. Following addition of phosphate to the oxidised protein, the corresponding spectrum shows partial hyperfine structure consistent with formation of prFMN semiquinone. Inset: UV–Vis of the EPR samples. Stereo images of **a**–**d** are available in Supplementary Figs. 6–9

unrelated prenyltransferases[27,28]. In the DMAP-specific *Pa*UbiX, the conserved E140 is proposed to assist with formation of the phosphate-leaving group through protonation. The crystal structure of the DMAPP-specific *An*UbiX reveals a similar interaction between E162 and the DMAPP alpha-phosphate is established, as part of a highly conserved network of polar interaction. In fact, we show DMAPP-specific UbiX enzymes such as *An*UbiX and *Ec*UbiX appear to catalyse the prenyltransferase step with DMAP, but in this case do not proceed to the cyclisation step. This suggests the pyrophosphate-leaving

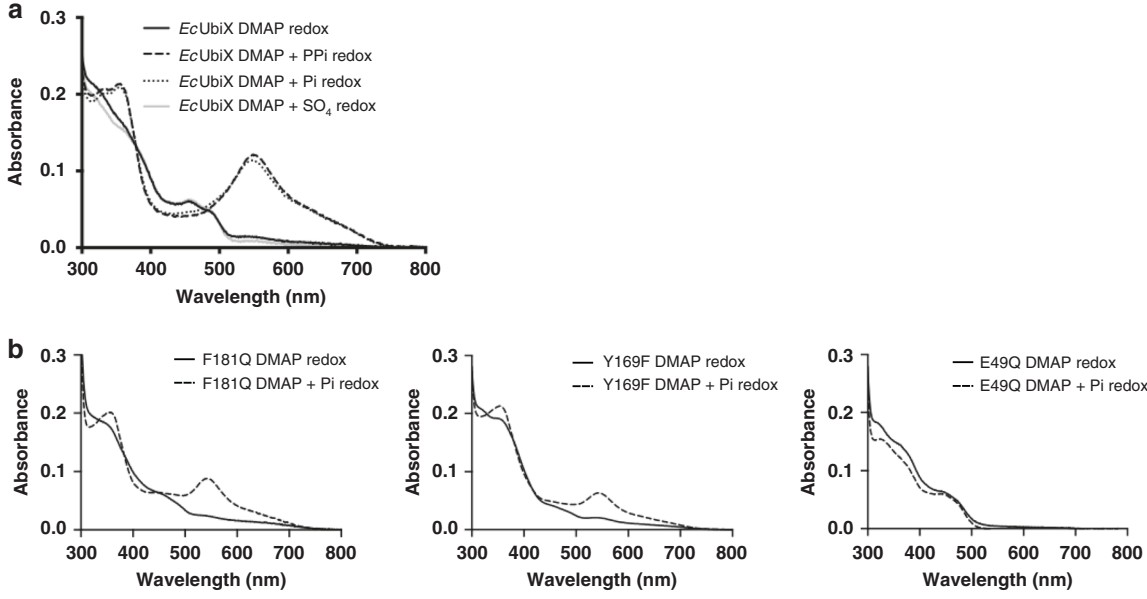

**Fig. 5** Phosphate salvages prFMN formation with DMAP in DMAPP-specific UbiX. **a** UV–Vis spectra following oxidation of FMNH$_2$:EcUbiX incubated with DMAP and 5 mM pyrophosphate, phosphate or sulphate to the reduced reaction. Only the addition of either phosphate and pyrophosphate salvages prFMNH$_2$ production as indicated by purple prFMN semiquinone formation. **b** UV–Vis spectra following oxidation of FMNH$_2$:F181Q, Y169F and E49Q PaUbiX variants with and without the addition of 5 mM phosphate to the reduced reaction. Both F181Q and Y169F variants form prFMNH$_2$ when phosphate is added, while E49Q remains unaffected by phosphate addition and unable to yield the cofactor

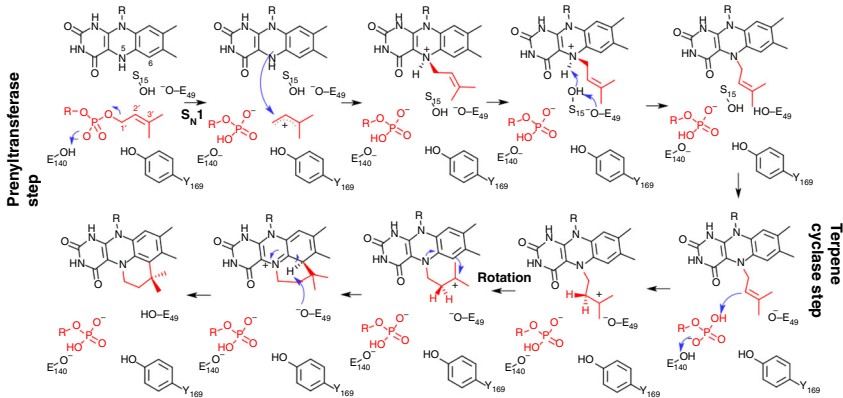

**Fig. 6** A detailed mechanism for UbiX. The prFMNH$_2$ formation in UbiX occurs through sequential formation of two FMN–dimethylallyl linkages, the N5–C1' and C6–C3' bonds. The N5-prenylation resembles other unrelated prenyltransferases, requiring the dimethylallyl moiety to allow formation of a resonance stabilised carbocation via an S$_N$1 mechanism using either DMAP or DMAPP as substrate. Flavin reduction is key to initiation of the N5–C1' bond-formation step. Following the prenyltranserase step, the Friedel-Crafts alkylation of the flavin C6 is initiated by proton transfer via the bound (pyro)phosphate, requiring the presence of the conserved E49 to allow formation of the prFMNH$_2$ product by proton abstraction from C6

group does not play a key role in the N5-prenylation, but the retention of the leaving group is crucial in the C6–C3' bond-formation process in these enzymes. However, the O157UbiX enzyme is able to use both DMAP and DMAPP (albeit with different efficiency) to form the prFMNH$_2$ product, suggesting the pyrophosphate beta-phosphate group does not play a key role in formation of either of the flavin–dimethylallyl linkages.

Despite the relatively modest changes that occur between DMAPP-specific AnUbiX and DMAP-specific PaUbiX attempts to bestow DMAPP activity on the latter by mutation of F181 were unsuccessful. Instead, the F181Q mutation in particular affected the ability of PaUbiX to complete the catalytic cycle and resulted in formation of stalled N5-prenylated species. A similar behaviour was previously observed with PaUbiX Y169F and E49Q variants,

and mimics the DMAP reaction of AnUbiX/ EcUbiX. This suggests the phosphate environment is key to catalysis of the C6–C3' linkage formation, proposed to occur via tertiary carbocation formation through phosphate-mediated protonation. It is possible the pK$_a$ and/or orientation of the bound phosphate are sufficiently perturbed in the PaUbiX variants to hinder this step. Alternatively, affinity for the phosphate might be reduced to a level where the phosphate prematurely leaves the active site prior to cyclisation. The fact that addition of phosphate (but not sulphate) to the reaction buffer salvages prFMNH$_2$ formation for both Y169F and F181Q PaUbiX variants, strongly suggest that retention of the phosphate is predominantly affected and confirms phosphate plays a key role in the cyclisation process. Furthermore, the addition of phosphate is also able to salvage

prFMNH$_2$ formation for the reaction of the $Ec$UbiX with DMAP, confirming the pyrophosphate group is not required in a strict sense for catalysis. Hence, our data confirm direct involvement of the phosphate/pyrophosphate alpha-phosphate group in the cyclisation step, something that has also been proposed in class I terpenoid synthases. Indeed, multistep general acid–base catalysis by the inorganic pyrophosphate coproduct has been suggested to be prevalent in terpenoid cyclases[24,25,28–30]. In the case of UbiX, the high affinity for the N5-prenylated intermediate has allowed us to directly demonstrate involvement of inorganic (pyro) phosphate coproduct in the cyclisation step in selected variants. Our data demonstrate mutation can result in premature loss of the essential inorganic (pyro)phosphate, but that this can be remedied through an increase in inorganic (pyro)phosphate concentration. Similar issues are likely to affect efforts at reprogramming terpenoid synthases to generate novel products, although addition of (pyro)phosphate may not always be sufficient for restoration of catalysis[31].

## Methods

**Cloning and mutagenesis.** The gene encoding $An$UbiX (XP_001390532) was codon optimised and synthesised (GenScript). The gene was amplified by PCR with the primers $An$UbiX-F and $An$UbiX-R using Phusion polymerase (NEB). The PCR product was cloned using In-Fusion (Clontech) into pET21b (Merck) that had been linearised with $Nde$I/$Xho$I (NEB).

O157:H7 $E. coli$ $ubiX$ (WP_000767724) was synthesised (GenScript) with an N terminal 6x His tag. The gene was amplified by PCR using the primers $O157$UbiX-F and $O157$UbiX-R, and the product was cloned into pET21b using In-Fusion (Clontech) that had been linearised using $Nde$I and $Xho$I.

$E. coli$ K12 $ubiX$ was cloned from genomic material. The gene was amplified by PCR using $Ec$UbiX-F and $Ec$UbiX-R primers and cloned into pNic28-Bsa4[32] cut with $Bsa$I using In-Fusion (Clontech).

$Pa$UbiX was previously synthesised (GenScript), codon optimised for $E. coli$. The gene was amplified with Phusion polymerase using the primers $Pa$UbiX-F and $Pa$UbiX-R, and cloned into pNic28-Bsa4, cut with $Nco$I and $Hin$dIII, using In-Fusion. Previously described $Pa$UbiX variants were used in this study, generated using QuikChange kit (Agilent), along with $Pa$UbiX-W200H. QuikChange variants were generated using primers denoted $Pa$UbiX- < variant > -QCF/R.

$Pa$UbiX-F181Q and $Pa$UbiX-F181H were generated using Q5 mutagenesis kit (NEB) using primers denoted either F181H/Q-Q5F along with F181X-Q5R as described below, designed with the NEBaseChanger web tools.

The sequence of all plasmids was confirmed using Eurofins Genomics sequencing service and BL21 (DE3) (NEB) were transformed.

Primers used in this study:

$An$UbiX-F; AAGGAGATATACATATGTTTAACTCTCTGCTGTC
$An$UbiX-R; GGTGGTGGTGCTCGAGTCATTTTTCCCAGC
$O157$UbiX-F; AAGGAGATATACATATGATGGGTTCGTCGCACCAC
$O157$UbiX-R;
GGTGGTGGTGCTCGAGTTATTCATTTTCCTGTGAGAAGTTG
$Ec$UbiX-F; TACTTCCAATCCATGAAACGACTCATTGTAGGC
$Ec$UbiX-R; TATCCACCTTTACTGTTATGCGCCCTGCCA
$Pa$UbiX-F; TACTTCCAATCCATGAGCGGTCCGGAAC
$Pa$UbiX-R GTGCGGCCGCAAGCTTTTATTCGTCTGAAAC
$Pa$UbiX-F181Q-Q5F; TCTGGTCGACCAAGTGGTTGCGCGTATTCTG
$Pa$UbiX-F181H-Q5F; TCTGGTCGACCACGTGGTTGCGC
$Pa$UbiX-F181X-Q5R; TCTTCCACGCTCTGCGGC
$Pa$UbiX-W200H-QCF;
GGATATGCTGCCGCGCCCATGGGGAACAGCACCTTG
$Pa$UbiX-W200H-QCR;
CAAGGTGCTGTTCCCCATGGCGCGGCAGCATATCC
$Pa$UbiX-Y169F-QCF, GGCTGCACCGGGTTTTTTCCACCAGCC
$Pa$UbiX-Y169F-QCR; GGCTGGTGGAAAAAACCCGGTGCAGCC
$Pa$UbiX-E49Q-QCF; CTGGTGATGGCCACCCAGACGGATGTTGCTCTG
$Pa$UbiX-E49Q-QCR; CAGAGCAACATCCGTCTGGGTGGCCATCACCAG

**Expression.** BL21(DE3) cells were grown at 37 °C in TB supplemented with 50 μg mL$^{-1}$ antibiotics to an OD$_{600}$ ~0.6. Gene expression was induced by the addition of 0.2 mM IPTG and the cultures grown overnight, typically for 18 h, at 18 °C. Cells were harvested and stored at −20 °C.

**Purification.** Cells expressing $An$UbiX were thawed and resuspended in buffer A (200 mM NaCl, 50 mM Tris-HCl pH 7.5) supplemented with 1 mM FMN, RNase, DNase and complete EDTA-free Protease inhibitor cocktail (Sigma). Once homogenous cells were lysed on ice using a Bandolin sonicator, at 30% amplitude with 15 × 20 s bursts interspersed with 40 s intervals. The lysate was clarified by

centrifugation at 125,000 $g$ for 60 min (4 °C). The supernatant was applied to a Ni-NTA agarose column (Qiagen). The column washed with three column volumes of buffer A supplemented with 10 mM imidazole followed by three column volumes of buffer supplemented with 40 mM imidazole. The protein was eluted with buffer A supplemented with 500 mM imidazole. Samples were subjected to SDS–PAGE analysis, and fractions found to contain the purified $An$UbiX were pooled and concentrated using a 10 kD MWCO Vivaspin centrifugal device (Sartorius).

$Pa$UbiX cells were resuspended in 50 mM Tris-HCl, 500 mM NaCl, pH 8 (buffer A), supplemented with 1 mM FMN, RNase, DNase and complete EDTA-free protease inhibitor cocktail (Sigma). Once homogenised cells were lysed using a Constant Systems Cell Disruptor at 20 kPsi. The lysate was clarified by centrifugation at 125,000 $g$ for 60 min (4 °C) prior to loading on a gravity flow Ni-NTA agarose column (Qiagen). The column was washed and the protein eluted in the same manner as described for $An$UbiX. The eluted protein was desalted using 10-DG column (Biorad) into 20 mM Tris-HCl, 200 mM NaCl, pH 8 for storage and further experiments. $Ec$UbiX was purified as per the protocol used for $Pa$UbiX, with the addition of 10% glycerol to the purification buffers.

$O157$UbiX was purified as previously described;[26] cells were resuspended in lysis buffer containing 50 mM Tris-HCl (pH 7.5), 400 mM NaCl, 20 mM imidazole, 5% (v/v) glycerol, 10 mM β-mercaptoethanol, 1× BugBuster solution (Novagen), supplemented with complete EDTA-free Protease inhibitor cocktail, lysozyme, RNase and DNase. The lysate was clarified by centrifugation at 125,000 $g$ for 60 min (4 °C), the supernatant was incubated with 2 mL DEAE–Sepharose resin for 30 min. The unbound fraction was applied to Ni-NTA agarose (Qiagen), before washing with 50 mM Tris-HCl (pH 7.5), 400 mM NaCl, 40 mM imidazole, 5% (v/v) glycerol, 10 mM β-mercaptoethanol, prior to elution with buffer containing 200 mM imidazole. The protein was concentrated with the addition of 1 mM FMN, prior to desalting using 10-DG column (Biorad) into 50 mM Tris-HCl (pH 7.5), 400 mM NaCl, 5% (v/v) glycerol, 5 mM DTT for storage. Protein was desalted into 20 mM Tris, 200 mM NaCl (pH 8) for the experiments.

**Chemical syntheses.** Geraniol, 3-methylbutanol and pentanol (1 mmol) were separately subjected to phosphorylation with ditriethylammonium phosphate salt (2.4 mmol) and trichloroacetonitrile (6 mmol) in acetonitrile (20 mL)[33]. This predominantly gives monophosphorylation to the corresponding derivatives 62–75% yield, with ~15% of the pyrophosphorylated derivatives also observed. The monophosphorylated derivatives were isolated by column chromatography (silica gel, $n$-propyl alcohol/concentrated ammonia, 6/3, v/v). The isolated products were lyophilised to a white powder. $^1$H, and $^{31}$P NMR were found to be in accordance with the literature.

Geranyl monophosphate (Supplementary Fig. 11) $^1$H NMR (400 MHz, D$_2$O) δ 5.42–5.39 (m, 1 H), 5.20–5.17 (m, 1 H), 4.39–4.36 (m, 1 H), 2.14–2.05 (m, 4 H), 1.68 (s, 3 H), 1.67 (s, 3 H), 1.60 (s, 3 H) $^{31}$P NMR (162 MHz, D$_2$O) δ 2.02.

3-Methylbutanyl monophosphate (Supplementary Fig. 12): $^1$H NMR (400 MHz, D$_2$O) δ 3.72 (q, $J = 6.7$, 6.8 Hz, 2 H), 1.63–1.50 (m, 1 H), 1.37 (q, $J = 7.0$ Hz, 2 H), 0.81–0.74 (m, $J = 6.6$, 2.9 Hz, 6 H) $^{31}$P NMR (162 MHz, D$_2$O) δ 1.85 (t, $^3J$$^{31}$P-$^1$H $= 5.7$ Hz).

Pentanyl monophosphate (Supplementary Fig. 13): $^1$H NMR (400 MHz, D$_2$O) δ 3.69–3.64 (q, $J = 6.7$ Hz, 2 H), 1.55–1.44 (m, 2 H), 1.22–1.17 (dq, $J = 7.2$, 3.6 Hz, 4 H), 0.78–0.73 (m, 6 H) $^{31}$P NMR (162 MHz, D$_2$O) δ 1.87.

**Crystallography.** $An$UbiX crystals were grown in JCSG + screen (Molecular Dimensions) condition D3 (0.2 M NaCl, 0.1 M Na/K phosphate pH 6.2, 30% v/v PEG200). Crystals were flash cooled in liquid nitrogen.

Wild-type $Pa$UbiX crystals were grown in SG1 screen (Molecular Dimensions) condition F4 (1 M sodium citrate tribasic dihydrate, 0.1 M sodium cacodylate, pH 6.5). The crystals were subjected to soaking with FMN and IMP and were reduced with sodium dithionite, and were cryoprotected in mineral oil before flash cooling in liquid nitrogen.

F181Q crystals grew in multiple conditions. Crystals from D7 LMB screen (Molecular Dimensions) (15% w/v PEG 3350, 0.1 M MES pH 6.2) were soaked with FMN and DMAP, and redox cycled with sodium dithionite prior to cryoprotection in 20% PEG 200 in mother liquor and flash cooling.

F181H crystals grew in multiple conditions. Crystals from F2 LMB screen (Molecular Dimensions) (16% w/v PEG 4000, 20% v/v glycerol, 0.1 M sodium citrate pH 5.8, 0.1 M ammonium sulphate) were soaked with FMN and DMAP and redox cycled with sodium dithionite prior flash cooling in liquid nitrogen.

W200H crystals were produced in various conditions of the PGA screen (Molecular Dimensions). Crystals in condition C3 (0.1 M ammonium sulphate, 0.3 M sodium formate, 0.1 M sodium acetate, pH 5.0, 3% (w/v) gamma—polyglutamic acid (Na$^+$ form, LM), 20% MPD) were soaked with geranyl monophosphate and subjected to reduction with sodium dithionite prior to flash cooling in liquid nitrogen.

$An$UbiX crystal data were collected at Diamond Light Source on proposal MX-8997 on beamline i03, W200H $Pa$UbiX crystal data were collected on proposal MX-12788 on beamline i03. Other data for $Pa$UbiX crystals were collected at Diamond Light Source on proposal MX-17773 on beamline i04, with the exception of F181H-DMAP (oxidised) which was collected on i24.

Data were processed using the xia2 DIALS autoprocessing pipeline at Diamond Light Source[34,35]. For W200H $Pa$UbiX, two isomorphous crystal datasets were

merged using BLEND[36]. Structures were solved using molecular replacement with the model 4ZAF using Phaser MR[37]. Models were refined using phenix.refine[38] or REFMAC5[39] in the Phenix and CCP4 suites, respectively[40,41]. Models were iteratively built manually using Coot[42], and ligand libraries defined using phenix. eLBOW[43] and AceDRG[44]. Omit maps shown are Polder maps[45]. Data were deposited in the PDB under the accession codes 6QLG, 6QLH, 6QLI, 6QLJ, 6QLK, 6QLL and 6QLV. Crystallographic data collection and refinement statistics are shown in Supplementary Table 1.

**UV–Vis spectroscopy**. UV-Vis absorbance spectra were recorded with a Cary UV–Vis spectrophotometer. The protein concentrations were estimated using calculated extinction coefficients calculated using a ExPASY ProtParam tool. The extinction coefficients used were: $An$UbiX $\varepsilon_{280} = 22460\,M^{-1}\,cm^{-1}$, $Pa$UbiX and $O157$UbiX $\varepsilon_{280} = 16960\,M^{-1}\,cm^{-1}$, Y169F $Pa$UbiX $\varepsilon_{280} = 13980\,M^{-1}\,cm^{-1}$, $Ec$UbiX and W200H $Pa$UbiX $\varepsilon_{280} = 11460\,M^{-1}\,cm^{-1}$.

**Single turnover assays**. Single turnover assays were performed with protein which was fully occupied by FMN; to ensure full occupancy, after purification proteins were incubated with 1 mM FMN and were desalted into 20 mM Tris-HCl, 200 mM NaCl, pH 8 using a 10-DG desalting column (Biorad). Assays were performed in a Belle Technology glovebox in an anoxic nitrogen environment. Protein concentrations were adjusted to ~200 μM, and were reduced by the titration of minimal volumes of sodium dithionite. Reduction was followed by the decrease in absorbance at 446 nm. Reactions were performed with 2 mM DMAP(P) and where added 5 mM Na(P)Pi or Na₂SO₄, pH 8 was used.

**Binding assays**. The binding of alkyl-phosphate compounds was measured using 50 μM $Pa$UbiX, with the addition of 500 μM compound. The spectra were measured before and after the addition of compounds, and difference spectra were calculated by the subtraction of the unbound spectrum from the bound spectrum.

**EPR spectroscopy**. EPR spectra were obtained using a Bruker E500/580 EPR spectrometer. Continuous wave X-band (~9.4 GHz) EPR spectra employed a Bruker "Super High Q" cavity (ER 4122SHQE) coupled to an Oxford Instruments ESR900 helium flow cryostat for temperature control. Spectra were acquired at 20 K using 10-microwatt microwave power, 100-kHz field modulation frequency, and 1 -G modulation amplitude. Samples were stored as 300 μL aliquots in 4 -mm quartz tubes (Wilmad) prior to recording spectra.

**Reporting summary**. Further information on research design is available in the Nature Research Reporting Summary linked to this article.

## Data availability

Crystallography data have been deposited in the PDB, under the accession codes 6QLG, 6QLH, 6QLI, 6QLJ, 6QLK, 6QLL and 6QLV. A reporting summary for this Article is available as a Supplementary Information file. All other data supporting the findings of this study are available from the corresponding author on reasonable request.

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

## Acknowledgements

This work was supported by BBSRC grants BB/K017802/1 and BB/P000622/1 and the ERC grant ADG_695013. D.L. is a Royal Society Wolfson Merit Award holder. We thank Diamond Light Source (Didcot, UK) for beamline access under the proposals MX-8997, MX-12788 and MX-17773. We would like to acknowledge Manchester Protein Structure Facility for assistance.

## Author contributions

S.A.M. produced F181 and W200H *Pa*UbiX mutants and cloned *Ec*UbiX, performed all UV–Vis and crystallography (with the exception of *An*UbiX) experiments, and analysed all data, K.A.P.P. produced and crystallised *An*UbiX. K.F. assisted in the production of EPR samples, collected and analysed EPR data with S.E.J.R. M.D.W. generated E49Q and Y169F *Pa*UbiX mutants and cloned *O157*UbiX. A.N.C. synthesised monophosphorylated compounds. A.B. assisted in collection of UV–Vis data. D.L. conceived and coordinated the study. S.A.M. wrote the paper with D.L., all authors approved the paper.

## Additional information

**Competing interests:** The authors declare no competing interests.

