## [Peer Review File · Nature Communications]

Reviewers' comments:

Reviewer #1 (Remarks to the Author):

This manuscript describes detailed structural and some mechanistic investigation of the flavin prenyltransferase UbiX. The results are reasonably well described (although see below), and clearly offer insight into the underlying catalytic mechanism. However, the context for the work is not well presented. Although UbiX seems to be widely dispersed (at least in microbes), its importance is not noted (e.g., in metabolism, for pathogen virulence, etc). This might help with the visibility of the work. The results are solid and provide insight, but need a bit more discussion and some clarification. First, despite the argument (supported by the chemical complementation data) that (pyro)phosphate retention is required for the second slower C6-alkylation step, which depends to some extent on the phenylalanine (F181 in PaUbiX) targeted here by mutagenesis, the crystal structure of the F181Q mutant following redox cycling to generate the initial N5-alkylation product still contains the phosphate co-product. This counterpoint needs to be mentioned and discussed (even if just to argue that this retention is a result of crystal rigidity). Second, the discussion is not quite correct regarding the role of the (pyro)phosphate (line 296). In particular, as shown here, the allylic (pyro)phosphate is required as a leaving group for initial N5-alkylation. What the authors presumably meant to say is that retention of the (pyro)phosphate co-product is required for the subsequent C6-alkylation (i.e., C6-C3' bond formation). This crucial distinction must be made. In addition, while the authors suggest that the insightful chemical complementation approach taken here should be taken in terpenoid synthases, this has been previously attempted (albeit unsuccessfully – e.g., with abietadiene synthase [Peters et al., 2000, *Biochemistry* 39:15592]). Finally, Figure 6 needs to be corrected and clarified (the final product is missing a double-bond in what should be the aromatic “right-hand” ring, the penultimate intermediate should have a positive charge indicated as it is an iminium, and it might be helpful to indicate the double-bond rotations in the preceding intermediate as well).

Reviewer #2 (Remarks to the Author):

This manuscript drills down into the mechanistic details of UbiX catalysis and the preference for DMAP vs. DMAPP in homologs. The N5-C1' bond formation appears to be an SN1 mechanism as observed in other prenyltransferases, while the C6-C3' bond formation appears similar to the class I terpene cyclase mechanism that is initiated by (pyro)phosphate. The chemistry is interesting and the experiments employ a large number of complementary techniques; activity, site-directed mutagenesis, UV-vis and EPR spectroscopy, and crystallography. The quality of the data are excellent and the mechanistic interpretation logical and persuasive. However, the authors do not make a strong case for why this should be published in a broad readership journal: ubiquinone, for example,

is not mentioned anywhere in the article. Why should a reader who is not an enzymologist/chemical biologist care about UbiX? Even if the “big picture” is included, I would suggest that only an enzymologist/organic chemist/chemical biologist could follow and appreciate the elegance of the experiments and the minutiae of the UbiX mechanism. As such the article appears more suited to a specialist journal.

I also found it hard to understand the spatial and structural arguments based on the figures included in the manuscript. There are so many residues and even with the two orientation image of Fig. 2c I could not clearly envisage the arrangement. This is particularly important as so many of the mechanistic arguments relate to the juxtaposition of atoms observed in the structures. Even if there is no space for stereo figures in the main paper, stereo images of (minimally) Fig. 2b, 2c, 3b, 4b, 4d should be placed in the SI. I could not make out the fit of the model within the electron density for Fig. 4b in particular. In Fig. 4 change the dimethylallyl coloring to be different from the isoalloxazine moiety to make it easier to distinguish the transferred atoms. None of the structural figures contain E49 – a key mutation site in the study and a key player in the mechanism. As such I could not fully check and follow the structural mechanistic arguments that are based on atom juxtapositions in the manuscript.

Minor comments:

Line 116. I think the authors mean to reference Fig 2c here not Fig. 2d.

Fig. 2c. Describe the rotational relationship between the two panels in Fig. 2c in the legend.

Please include an SI Fig. containing chemical structures of all the DMAP(P) compounds used in the study.

Line 235. The reference to Fig. 4h should be moved up to line 229.

Fig. 6. It would be helpful to add the key atom numbers again in the first panel of Fig. 6 as in Fig. 1.

Line 320. “data confirms” should be “data confirm”

Line 327. "data demonstrates" should be "data demonstrate"

Line 481. "with a the omit" should be "with the omit"

Line 497. "previously" is misspelt in the legend for Fig. 3 on page 7.

Lines 508 and 510. "Crystal structure the" should be "Crystal structure of the"

Line 532. "semiquinone" is misspelt in the legend for Fig. 5 on page 11.

Carrie Wilmot (University of Minnesota)

Reviewer #3 (Remarks to the Author):

The present study uses x-ray structural and spectroscopic data to posit a two-step mechanism for the prenylation of FMNH₂ carried out by UbiX enzymes. First the prenyl group is transferred from a molecule of the cosubstrate DMAP or DMAPP forming a N5-C1' linkage while losing the (pyro)phosphate group. Which substrate is used seems to depend on whether there is sufficient space for the pyrophosphate group in the substrate binding site. The second step forms a bond between C6 of FMNH₂ and C3' of the pendant prenyl group making a new non-aromatic ring. Interestingly, this second step requires the presence of the phosphate leaving group from the first step. This is shown unequivocally in studies in which exogenous phosphate is used to drive the reaction is used to drive the reaction to completion. EPR spectroscopy is used to show that in a mutant (F181Q) which is unable to complete the second step, can do so in the presence of inorganic phosphate.

I can find no fault in any of the conclusions derived from this study. My only criticism is that it seems all the mechanistic elements described in this finding were hypothesized before in other studies including those by the authors.

Point-by-point response to reviewer comments made.

This manuscript describes detailed structural and some mechanistic investigation of the flavin prenyltransferase UbiX. The results are reasonably well described (although see below), and clearly offer insight into the underlying catalytic mechanism. However, the context for the work is not well presented.

Although UbiX seems to be widely dispersed (at least in microbes), its importance is not noted (e.g., in metabolism, for pathogen virulence, etc). This might help with the visibility of the work.

We have improved the introduction to state the diversity of systems which UbiX is involved in, and the potential application of the UbiD-UbiX system, for which understanding of UbiX is vital.

The results are solid and provide insight, but need a bit more discussion and some clarification. First, despite the argument (supported by the chemical complementation data) that (pyro)phosphate retention is required for the second slower C6-alkylation step, which depends to some extent on the phenylalanine (F181 in PaUbiX) targeted here by mutagenesis, the crystal structure of the F181Q mutant following redox cycling to generate the initial N5-alkylation product still contains the phosphate co-product. This counterpoint needs to be mentioned and discussed (even if just to argue that this retention is a result of crystal rigidity).

We have included a statement on page 12-13 which addresses this issue. We are unable to confirm the reasons for this observation, but have postulated potential reasons.

Second, the discussion is not quite correct regarding the role of the (pyro)phosphate (line 296). In particular, as shown here, the allylic (pyro)phosphate is required as a leaving group for initial N5-alkylation. What the authors presumably meant to say is that retention of the (pyro)phosphate co-product is required for the subsequent C6-alkylation (i.e., C6-C3' bond formation). This crucial distinction must be made.

We have clarified our meaning, stating that the leaving group retention is required.

In addition, while the authors suggest that the insightful chemical complementation approach taken here should be taken in terpenoid synthases, this has been previously attempted (albeit unsuccessfully – e.g., with abietadiene synthase [Peters et al., 2000, Biochemistry 39:15592]).

We have noted that the addition of (pyro)phosphate may not be enough for restoration of catalysis in all systems.

Finally, Figure 6 needs to be corrected and clarified (the final product is missing a double-bond in what should be the aromatic “right-hand” ring, the penultimate intermediate should have a positive charge indicated as it is an iminium, and it might be helpful to indicate the double-bond rotations in the preceding intermediate as well).

We have made the requested changes to Fig 6.

Reviewer #2 (Remarks to the Author):

This manuscript drills down into the mechanistic details of UbiX catalysis and the preference for DMAP vs. DMAPP in homologs. The N5-C1' bond formation appears to be an SN1 mechanism as observed in other prenyltransferases, while the C6-C3' bond formation appears similar to the class I terpene cyclase mechanism that is initiated by (pyro)phosphate. The chemistry is interesting and the experiments employ a large number of complementary techniques; activity, site-directed mutagenesis, UV-vis and EPR spectroscopy, and crystallography. The quality of the data are excellent and the mechanistic interpretation logical and persuasive. However, the authors do not make a strong case for why this should be published in a broad readership journal: ubiquinone, for example, is not mentioned anywhere in the article. Why should a reader who is not an enzymologist/chemical biologist care about UbiX? Even if the "big picture" is included, I would suggest that only an enzymologist/organic chemist/chemical biologist could follow and appreciate the elegance of the experiments and the minutiae of the UbiX mechanism. As such the article appears more suited to a specialist journal.

We have addressed the issue of the. "big picture" by highlighting the uses of the UbiD-UbiX system, both within Nature and for the biotechnological uses which it could afford (see also response to first comment of reviewer 1)

I also found it hard to understand the spatial and structural arguments based on the figures included in the manuscript. There are so many residues and even with the two orientation image of Fig. 2c I could not clearly envisage the arrangement.

This is particularly important as so many of the mechanistic arguments relate to the juxtaposition of atoms observed in the structures. Even if there is no space for stereo figures in the main paper, stereo images of (minimally) Fig. 2b, 2c, 3b, 4b, 4d should be placed in the SI.

Stereo images have been made as requested and placed in supplementary figures.

I could not make out the fit of the model within the electron density for Fig. 4b in particular. In Fig. 4 change the dimethylallyl coloring to be different from the isoalloxazine moiety to make it easier to distinguish the transferred atoms.

All crystal figures showing density (with the exception of Fig 3b where the molecules are coloured according to DMAP/IMP) have been improved by changing the colour of DMAP(P) or DMAP(P) derived adducts to green to contrast from the magenta FMN. As such we have also changed the electron density to grey, and reduced the mesh width appearance to improve clarity.

None of the structural figures contain E49 – a key mutation site in the study and a key player in the mechanism. As such I could not fully check and follow the structural mechanistic arguments that are based on atom juxtapositions in the manuscript. This has been corrected, and figures now contain this essential residue

Minor comments:

Line 116. I think the authors mean to reference Fig 2c here not Fig. 2d.

Fig. 2c. Describe the rotational relationship between the two panels in Fig. 2c in the legend.

Please include an SI Fig. containing chemical structures of all the DMAP(P) compounds used in the study.

Line 235. The reference to Fig. 4h should be moved up to line 229.

Fig. 6. It would be helpful to add the key atom numbers again in the first panel of Fig. 6 as in Fig. 1.

Line 320. "data confirms" should be "data confirm"

Line 327. "data demonstrates" should be "data demonstrate"

Line 481. "with a the omit" should be "with the omit"

Line 497. "previously" is misspelt in the legend for Fig. 3 on page 7.

Lines 508 and 510. "Crystal structure the" should be "Crystal structure of the"

Line 532. "semiquinone" is misspelt in the legend for Fig. 5 on page 11.

All minor comments have been addressed.

Reviewer #3 (Remarks to the Author):

The present study uses x-ray structural and spectroscopic data to posit a two-step mechanism for the prenylation of FMNH₂ carried out by UbiX enzymes. First the prenyl group is transferred from a molecule of the cosubstrate DMAP or DMAPP forming a N5-C1' linkage while losing the (pyro)phosphate group. Which substrate is used seems to depend on whether there is sufficient space for the pyrophosphate group in the substrate binding site. The second step forms a bond between C6 of FMNH₂ and C3' of the pendant prenyl group making a new non-aromatic ring. Interestingly, this second step requires the presence of the phosphate leaving group from the first step. This is shown unequivocally in studies in which exogenous phosphate is used to drive the reaction to completion. EPR spectroscopy is used to show that in a mutant (F181Q) which is unable to complete the second step, can do so in the presence of inorganic phosphate.

I can find no fault in any of the conclusions derived from this study. My only criticism is that it seems all the mechanistic elements described in this finding were hypothesized before in other studies including those by the authors.

No corrections required.

REVIEWERS' COMMENTS:

Reviewer #1 (Remarks to the Author):

This revised manuscript addresses all of my previously expressed concerns.